# Scheduled Sampling for Sequence Prediction with Recurrent Neural Networks

**Samy Bengio, Oriol Vinyals, Navdeep Jaitly, Noam Shazeer**
Google Research
Mountain View, CA, USA
{bengio,vinyals,ndjaitly,noam}@google.com

## Abstract

Recurrent Neural Networks can be trained to produce sequences of tokens given some input, as exemplified by recent results in machine translation and image captioning. The current approach to training them consists of maximizing the likelihood of each token in the sequence given the current (recurrent) state and the previous token. At inference, the unknown previous token is then replaced by a token generated by the model itself. This discrepancy between training and inference can yield errors that can accumulate quickly along the generated sequence. We propose a curriculum learning strategy to gently change the training process from a fully guided scheme using the true previous token, towards a less guided scheme which mostly uses the generated token instead. Experiments on several sequence prediction tasks show that this approach yields significant improvements. Moreover, it was used succesfully in our winning entry to the MSCOCO image captioning challenge, 2015.

## 1   Introduction

Recurrent neural networks can be used to process sequences, either as input, output or both. While they are known to be hard to train when there are long term dependencies in the data [1], some versions like the Long Short-Term Memory (LSTM) [2] are better suited for this. In fact, they have recently shown impressive performance in several sequence prediction problems including machine translation [3], contextual parsing [4], image captioning [5] and even video description [6].

In this paper, we consider the set of problems that attempt to generate a sequence of tokens of variable size, such as the problem of machine translation, where the goal is to translate a given sentence from a source language to a target language. We also consider problems in which the input is not necessarily a sequence, like the image captioning problem, where the goal is to generate a textual description of a given image.

In both cases, recurrent neural networks (or their variants like LSTMs) are generally trained to maximize the likelihood of generating the target sequence of tokens given the input. In practice, this is done by maximizing the likelihood of each target token given the current state of the model (which summarizes the input and the past output tokens) and the previous target token, which helps the model learn a kind of language model over target tokens. However, during inference, true *previous* target tokens are unavailable, and are thus replaced by tokens generated by the model itself, yielding a discrepancy between how the model is used at training and inference. This discrepancy can be mitigated by the use of a beam search heuristic maintaining several generated target sequences, but for continuous state space models like recurrent neural networks, there is no dynamic programming approach, so the effective number of sequences considered remains small, even with beam search.

The main problem is that mistakes made early in the sequence generation process are fed as input to the model and can be quickly amplified because the model might be in a part of the state space it has never seen at training time.

Here, we propose a *curriculum learning* approach [7] to gently bridge the gap between training and inference for sequence prediction tasks using recurrent neural networks. We propose to change the training process in order to gradually force the model to deal with its own mistakes, as it would have to during inference. Doing so, the model explores more during training and is thus more robust to correct its own mistakes at inference as it has learned to do so during training. We will show experimentally that this approach yields better performance on several sequence prediction tasks.

The paper is organized as follows: in Section 2, we present our proposed approach to better train sequence prediction tasks with recurrent neural networks; this is followed by Section 3 which draws links to some related approaches. We then present some experimental results in Section 4 and conclude in Section 5.

## 2 Proposed Approach

We are considering supervised tasks where the training set is given in terms of $N$ input/output pairs $\{X^i, Y^i\}_{i=1}^N$, where $X^i$ is the input and can be either static (like an image) or dynamic (like a sequence) while the target output $Y^i$ is a sequence $y_1^i, y_2^i, \ldots, y_{T_i}^i$ of a variable number of tokens that belong to a fixed known dictionary.

### 2.1 Model

Given a single input/output pair $(X, Y)$, the log probability $P(Y|X)$ can be computed as:

$$
\begin{aligned}
\log P(Y|X) &= \log P(y_1^T|X) \\
&= \sum_{t=1}^T \log P(y_t|y_1^{t-1}, X)
\end{aligned}
\tag{1}
$$

where $Y$ is a sequence of length $T$ represented by tokens $y_1, y_2, \ldots, y_T$. The latter term in the above equation is estimated by a recurrent neural network with parameters $\theta$ by introducing a state vector, $h_t$, that is a function of the previous state, $h_{t-1}$, and the previous output token, $y_{t-1}$, i.e.

$$
\log P(y_t|y_1^{t-1}, X; \theta) = \log P(y_t|h_t; \theta)
\tag{2}
$$

where $h_t$ is computed by a recurrent neural network as follows:

$$
h_t = \begin{cases} f(X; \theta) & \text{if } t = 1, \\ f(h_{t-1}, y_{t-1}; \theta) & \text{otherwise.} \end{cases}
\tag{3}
$$

$P(y_t|h_t; \theta)$ is often implemented as a linear projection[1] of the state vector $h_t$ into a vector of scores, one for each token of the output dictionary, followed by a softmax transformation to ensure the scores are properly normalized (positive and sum to 1). $f(h, y)$ is usually a non-linear function that combines the previous state and the previous output in order to produce the current state.

This means that the model focuses on learning to output the next token given the current state of the model AND the previous token. Thus, the model represents the probability distribution of sequences in the most general form - unlike Conditional Random Fields [8] and other models that assume independence between between outputs at different time steps, given latent variable states. The capacity of the model is only limited by the representational capacity of the recurrent and feedforward layers. LSTMs, with their ability to learn long range structure are especially well suited to this task and make it possible to learn rich distributions over sequences.

In order to learn variable length sequences, a special token, <EOS>, that signifies the end of a sequence is added to the dictionary and the model. During training, <EOS> is concatenated to the end of each sequence. During inference, the model generates tokens until it generates <EOS>.

## 2.2 Training

Training recurrent neural networks to solve such tasks is usually accomplished by using mini-batch stochastic gradient descent to look for a set of parameters $\theta^\star$ that maximizes the log likelihood of producing the correct target sequence $Y^i$ given the input data $X^i$ for all training pairs $(X^i, Y^i)$:

$$\theta^\star = \arg \max_\theta \sum_{(X^i, Y^i)} \log P(Y^i | X^i; \theta) . \qquad (4)$$

## 2.3 Inference

During inference the model can generate the full sequence $y_1^T$ given $X$ by generating one token at a time, and advancing time by one step. When an <EOS> token is generated, it signifies the end of the sequence. For this process, at time $t$, the model needs as input the output token $y_{t-1}$ from the last time step in order to produce $y_t$. Since we do not have access to the true previous token, we can instead either select the most likely one given our model, or sample according to it.

Searching for the sequence $Y$ with the highest probability given $X$ is too expensive because of the combinatorial growth in the number of sequences. Instead we use a beam searching procedure to generate $k$ "best" sequences. We do this by maintaining a heap of $m$ best candidate sequences. At each time step new candidates are generated by extending each candidate by one token and adding them to the heap. At the end of the step, the heap is re-pruned to only keep $m$ candidates. The beam searching is truncated when no new sequences are added, and $k$ best sequences are returned.

While beam search is often used for discrete state based models like Hidden Markov Models where dynamic programming can be used, it is harder to use efficiently for continuous state based models like recurrent neural networks, since there is no way to factor the followed state paths in a continuous space, and hence the actual number of candidates that can be kept during beam search decoding is very small.

In all these cases, if a wrong decision is taken at time $t - 1$, the model can be in a part of the state space that is very different from those visited from the training distribution and for which it doesn't know what to do. Worse, it can easily lead to cumulative bad decisions - a classic problem in sequential Gibbs sampling type approaches to sampling, where future samples can have no influence on the past.

## 2.4 Bridging the Gap with Scheduled Sampling

The main difference between training and inference for sequence prediction tasks when predicting token $y_t$ is whether we use the true previous token $y_{t-1}$ or an estimate $\hat{y}_{t-1}$ coming from the model itself.

We propose here a sampling mechanism that will randomly decide, during training, whether we use $y_{t-1}$ or $\hat{y}_{t-1}$. Assuming we use a mini-batch based stochastic gradient descent approach, for every token to predict $y_t \in Y$ of the $i^{th}$ mini-batch of the training algorithm, we propose to flip a coin and use the true previous token with probability $\epsilon_i$, or an estimate coming from the model itself with probability $(1 - \epsilon_i)^2$ The estimate of the model can be obtained by sampling a token according to the probability distribution modeled by $P(y_{t-1} | h_{t-1})$, or can be taken as the $\arg \max_s P(y_{t-1} = s | h_{t-1})$. This process is illustrated in Figure 1.

When $\epsilon_i = 1$, the model is trained exactly as before, while when $\epsilon_i = 0$ the model is trained in the same setting as inference. We propose here a *curriculum learning* strategy to go from one to the other: intuitively, at the beginning of training, sampling from the model would yield a random token since the model is not well trained, which could lead to very slow convergence, so selecting more often the true previous token should help; on the other hand, at the end of training, $\epsilon_i$ should favor sampling from the model more often, as this corresponds to the true inference situation, and one expects the model to already be good enough to handle it and sample reasonable tokens.

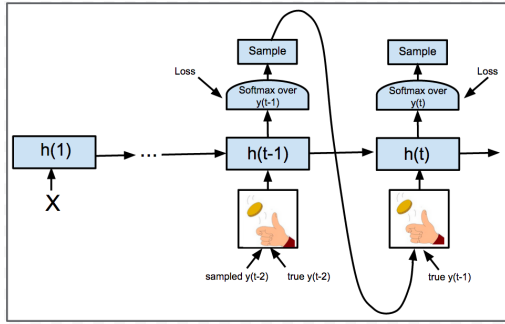

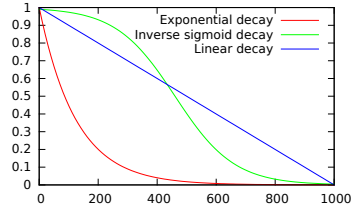

Figure 2: Examples of decay schedules.

Figure 1: Illustration of the Scheduled Sampling approach, where one flips a coin at every time step to decide to use the true previous token or one sampled from the model itself.

We thus propose to use a schedule to decrease $\epsilon_i$ as a function of $i$ itself, in a similar manner used to decrease the learning rate in most modern stochastic gradient descent approaches. Examples of such schedules can be seen in Figure 2 as follows:

- Linear decay: $\epsilon_i = \max(\epsilon, k - ci)$ where $0 \le \epsilon < 1$ is the minimum amount of truth to be given to the model and $k$ and $c$ provide the offset and slope of the decay, which depend on the expected speed of convergence.

- Exponential decay: $\epsilon_i = k^i$ where $k < 1$ is a constant that depends on the expected speed of convergence.

- Inverse sigmoid decay: $\epsilon_i = k/(k + \exp(i/k))$ where $k \ge 1$ depends on the expected speed of convergence.

We call our approach *Scheduled Sampling*. Note that when we sample the previous token $\hat{y}_{t-1}$ from the model itself while training, we could back-propagate the gradient of the losses at times $t \to T$ through that decision. This was not done in the experiments described in this paper and is left for future work.

## 3   Related Work

The discrepancy between the training and inference distributions has already been noticed in the literature, in particular for control and reinforcement learning tasks.

SEARN [9] was proposed to tackle problems where supervised training examples might be different from actual test examples when each example is made of a sequence of decisions, like acting in a complex environment where a few mistakes of the model early in the sequential decision process might compound and yield a very poor global performance. Their proposed approach involves a meta-algorithm where at each meta-iteration one trains a new model according to the current *policy* (essentially the expected decisions for each situation), applies it on a test set and modifies the next iteration policy in order to account for the previous decisions and errors. The new policy is thus a combination of the previous one and the actual behavior of the model.

In comparison to SEARN and related ideas [10, 11], our proposed approach is completely online: a single model is trained and the *policy* slowly evolves during training, instead of a batch approach, which makes it much faster to train[3] Furthermore, SEARN has been proposed in the context of reinforcement learning, while we consider the supervised learning setting trained using stochastic gradient descent on the overall objective.

Other approaches have considered the problem from a ranking perspective, in particular for parsing tasks [12] where the target output is a tree. In this case, the authors proposed to use a beam search both during training and inference, so that both phases are aligned. The training beam is used to find

the best current estimate of the model, which is compared to the guided solution (the truth) using a ranking loss. Unfortunately, this is not feasible when using a model like a recurrent neural network (which is now the state-of-the-art technique in many sequential tasks), as the state sequence cannot be factored easily (because it is a multi-dimensional continuous state) and thus beam search is hard to use efficiently at training time (as well as inference time, in fact).

Finally, [13] proposed an online algorithm for parsing problems that adapts the targets through the use of a *dynamic oracle* that takes into account the decisions of the model. The trained model is a perceptron and is thus not state-based like a recurrent neural network, and the probability of choosing the truth is fixed during training.

## 4    Experiments

We describe in this section experiments on three different tasks, in order to show that scheduled sampling can be helpful in different settings. We report results on image captioning, constituency parsing and speech recognition.

### 4.1    Image Captioning

Image captioning has attracted a lot of attention in the past year. The task can be formulated as a mapping of an image onto a sequence of words describing its content in some natural language, and most proposed approaches employ some form of recurrent network structure with simple decoding schemes [5, 6, 14, 15, 16]. A notable exception is the system proposed in [17], which does not directly optimize the log likelihood of the caption given the image, and instead proposes a pipelined approach.

Since an image can have many valid captions, the evaluation of this task is still an open problem. Some attempts have been made to design metrics that positively correlate with human evaluation [18], and a common set of tools have been published by the MSCOCO team [19].

We used the MSCOCO dataset from [19] to train our model. We trained on 75k images and report results on a separate development set of 5k additional images. Each image in the corpus has 5 different captions, so the training procedure picks one at random, creates a mini-batch of examples, and optimizes the objective function defined in (4). The image is preprocessed by a pretrained convolutional neural network (without the last classification layer) similar to the one described in [20], and the resulting image embedding is treated as if it was the first word from which the model starts generating language. The recurrent neural network generating words is an LSTM with one layer of 512 hidden units, and the input words are represented by embedding vectors of size 512. The number of words in the dictionary is 8857. We used an inverse sigmoid decay schedule for $\epsilon_i$ for the scheduled sampling approach.

Table 1 shows the results on various metrics on the development set. Each of these metrics is a variant of estimating the overlap between the obtained sequence of words and the target one. Since there were 5 target captions per image, the best result is always chosen. To the best of our knowledge, the baseline results are consistent (slightly better) with the current state-of-the-art on that task. While dropout helped in terms of log likelihood (as expected but not shown), it had a negative impact on the real metrics. On the other hand, scheduled sampling successfully trained a model more resilient to failures due to training and inference mismatch, which likely yielded higher quality captions according to all the metrics. Ensembling models also yielded better performance, both for the baseline and the schedule sampling approach. It is also interesting to note that a model trained while always sampling from itself (hence in a regime similar to inference), dubbed *Always Sampling* in the table, yielded very poor performance, as expected because the model has a hard time learning the task in that case. We also trained a model with scheduled sampling, but instead of sampling from the model, we sampled from a uniform distribution, in order to verify that it was important to build on the current model and that the performance boost was not just a simple form of regularization. We called this *Uniform Scheduled Sampling* and the results are better than the baseline, but not as good as our proposed approach. We also experimented with flipping the coin once per sequence instead of once per token, but the results were as poor as the *Always Sampling* approach.

Table 1: Various metrics (the higher the better) on the MSCOCO development set for the image captioning task.

| Approach vs Metric | BLEU-4 | METEOR | CIDER |
|---|---|---|---|
| Baseline | 28.8 | 24.2 | 89.5 |
| Baseline with Dropout | 28.1 | 23.9 | 87.0 |
| Always Sampling | 11.2 | 15.7 | 49.7 |
| Scheduled Sampling | **30.6** | **24.3** | **92.1** |
| Uniform Scheduled Sampling | 29.2 | 24.2 | 90.9 |
| Baseline ensemble of 10 | 30.7 | 25.1 | 95.7 |
| Scheduled Sampling ensemble of 5 | **32.3** | **25.4** | **98.7** |

It's worth noting that we used our scheduled sampling approach to participate in the 2015 MSCOCO image captioning challenge [21] and ranked first in the final leaderboard.

## 4.2 Constituency Parsing

Another less obvious connection with the *any-to-sequence* paradigm is constituency parsing. Recent work [4] has proposed an interpretation of a parse tree as a sequence of linear "operations" that build up the tree. This linearization procedure allowed them to train a model that can map a sentence onto its parse tree without any modification to the any-to-sequence formulation.

The trained model has one layer of 512 LSTM cells and words are represented by embedding vectors of size 512. We used an attention mechanism similar to the one described in [22] which helps, when considering the next output token to produce $y_t$, to focus on part of the input sequence only by applying a softmax over the LSTM state vectors corresponding to the input sequence. The input word dictionary contained around 90k words, while the target dictionary contained 128 symbols used to describe the tree. We used an inverse sigmoid decay schedule for $\epsilon_i$ in the scheduled sampling approach.

Parsing is quite different from image captioning as the function that one has to learn is almost deterministic. In contrast to an image having a large number of valid captions, most sentences have a unique parse tree (although some very difficult cases exist). Thus, the model operates almost deterministically, which can be seen by observing that the train and test perplexities are extremely low compared to image captioning (1.1 vs. 7).

This different operating regime makes for an interesting comparison, as one would not expect the baseline algorithm to make many mistakes. However, and as can be seen in Table 2, scheduled sampling has a positive effect which is additive to dropout. In this table we report the F1 score on the WSJ 22 development set [23]. We should also emphasize that there are only 40k training instances, so overfitting contributes largely to the performance of our system. Whether the effect of sampling during training helps with regard to overfitting or the training/inference mismatch is unclear, but the result is positive and additive with dropout. Once again, a model trained by always sampling from itself instead of using the groundtruth previous token as input yielded very bad results, in fact so bad that the resulting trees were often not valid trees (hence the "-" in the corresponding F1 metric).

Table 2: F1 score (the higher the better) on the validation set of the parsing task.

| Approach | F1 |
|---|---|
| Baseline LSTM | 86.54 |
| Baseline LSTM with Dropout | 87.0 |
| Always Sampling | - |
| Scheduled Sampling | **88.08** |
| Scheduled Sampling with Dropout | **88.68** |

## 4.3 Speech Recognition

For the speech recognition experiments, we used a slightly different setting from the rest of the paper. Each training example is an input/output pair $(X, Y)$, where $X$ is a sequence of $T$ input vectors $x_1, x_2, \cdots x_T$ and $Y$ is a sequence of $T$ tokens $y_1, y_2, \cdots y_T$ so each $y_t$ is aligned with the corresponding $x_t$. Here, $x_t$ are the acoustic features represented by log Mel filter bank spectra at frame $t$, and $y_t$ is the corresponding target. The targets used were HMM-state labels generated from a GMM-HMM recipe, using the Kaldi toolkit [24] but could very well have been phoneme labels. This setting is different from the other experiments in that the model we used is the following:

$$
\begin{aligned}
\log P(Y|X;\theta) &= \log P(y_1^T|x_1^T;\theta) \\
&= \sum_{t=1}^{T} \log P(y_t|y_1^{t-1}, x_1^t;\theta) \\
&= \sum_{t=1}^{T} \log P(y_t|h_t;\theta)
\end{aligned}
\tag{5}
$$

where $h_t$ is computed by a recurrent neural network as follows:

$$
h_t = \begin{cases}
f(\mathbf{o}_h, S, x_1;\theta) & \text{if } t = 1, \\
f(h_{t-1}, y_{t-1}, x_t;\theta) & \text{otherwise.}
\end{cases}
\tag{6}
$$

where $\mathbf{o}_h$ is a vector of 0's with same dimensionality as $h_t$'s and $S$ is an extra token added to the dictionary to represent the start of each sequence.

We generated data for these experiments using the TIMIT[4] corpus and the KALDI toolkit as described in [25]. Standard configurations were used for the experiments - 40 dimensional log Mel filter banks and their first and second order temporal derivatives were used as inputs to each frame. 180 dimensional targets were generated for each time frame using forced alignment to transcripts using a trained GMM-HMM system. The training, validation and test sets have 3696, 400 and 192 sequences respectively, and their average length was 304 frames. The validation set was used to choose the best epoch in training, and the model parameters from that epoch were used to evaluate the test set.

The trained models had two layers of 250 LSTM cells and a softmax layer, for each of five configurations - a baseline configuration where the ground truth was always fed to the model, a configuration (Always Sampling) where the model was only fed in its own predictions from the last time step, and three scheduled sampling configurations (Scheduled Sampling 1-3), where $\epsilon_i$ was ramped linearly from a maximum value to a minimum value over ten epochs and then kept constant at the final value. For each configuration, we trained 3 models and report average performance over them. Training of each model was done over frame targets from the GMM. The baseline configurations typically reached the best validation accuracy after approximately 14 epochs whereas the sampling models reached the best accuracy after approximately 9 epochs, after which the validation accuracy decreased. This is probably because the way we trained our models is not exact - it does not account for the gradient of the sampling probabilities from which we sampled our targets. Future effort at tackling this problem may further improve results.

Testing was done by finding the best sequence from beam search decoding (using a beam size of 10 beams) and computing the error rate over the sequences. We also report the next step error rate (where the model was fed in the ground truth to predict the class of the next frame) for each of the models on the validation set to summarize the performance of the models on the training objective. Table 3 shows a summary of the results

It can be seen that the baseline performs better next step prediction than the models that sample the tokens for input. This is to be expected, since the former has access to the groundtruth. However, it can be seen that the models that were trained with sampling perform better than the baseline during decoding. It can also be seen that for this problem, the "Always Sampling" model performs quite

well. We hypothesize that this has to do with the nature of the dataset. The HMM-aligned states have a lot of correlation - the same state appears as the target for several frames, and most of the states are constrained only to go to a subset of other states. Next step prediction with groundtruth labels on this task ends up paying disproportionate attention to the structure of the labels $(y_1^{t-1})$ and not enough to the acoustics input $(x_1^t)$. Thus it achieves very good next step prediction error when the groundtruth sequence is fed in with the acoustic information, but is not able to exploit the acoustic information sufficiently when the groundtruth sequence is not fed in. For this model the testing conditions are too far from the training condition for it to make good predictions. The model that is only fed its own prediction (Always Sampling) ends up exploiting all the information it can find in the acoustic signal, and effectively ignores its own predictions to influence the next step prediction. Thus at test time, it performs just as well as it does during training. A model such as the attention model of [26] which predicts phone sequences directly, instead of the highly redundant HMM state sequences, would not suffer from this problem because it would need to exploit both the acoustic signal and the language model sufficiently to make predictions. Nevertheless, even in this setting, adding scheduled sampling still helped to improve the decoding frame error rate.

Note that typically speech recognition experiments use HMMs to decode predictions from neural networks in a hybrid model. Here we avoid using an HMM altogether and hence we do not have the advantage of the smoothing that results from the HMM architecture and the language models. Thus the results are not directly comparable to the typical hybrid model results.

Table 3: Frame Error Rate (FER) on the speech recognition experiments. In next step prediction (reported on validation set) the ground truth is fed in to predict the next target like it is done during training. In decoding experiments (reported on test set), beam searching is done to find the best sequence. We report results on four different linear schedulings of sampling, where $\epsilon_i$ was ramped down linearly from $\epsilon_s$ to $\epsilon_e$. For the baseline, the model was only fed in the ground truth. See Section 4.3 for an analysis of the results.

| Approach | $\epsilon_s$ | $\epsilon_e$ | Next Step FER | Decoding FER |
|---|---|---|---|---|
| Always Sampling | 0 | 0 | 34.6 | 35.8 |
| Scheduled Sampling 1 | 0.25 | 0 | 34.3 | **34.5** |
| Scheduled Sampling 2 | 0.5 | 0 | 34.1 | 35.0 |
| Scheduled Sampling 3 | 0.9 | 0.5 | 19.8 | 42.0 |
| Baseline LSTM | 1 | 1 | 15.0 | 46.0 |

## 5  Conclusion

Using recurrent neural networks to predict sequences of tokens has many useful applications like machine translation and image description. However, the current approach to training them, predicting one token at a time, conditioned on the state and the previous correct token, is different from how we actually use them and thus is prone to the accumulation of errors along the decision paths. In this paper, we proposed a *curriculum learning* approach to slowly change the training objective from an easy task, where the previous token is known, to a realistic one, where it is provided by the model itself. Experiments on several sequence prediction tasks yield performance improvements, while not incurring longer training times. Future work includes back-propagating the errors through the sampling decisions, as well as exploring better sampling strategies including conditioning on some confidence measure from the model itself.

## Footnotes

[1] Although one could also use a multi-layered non-linear projection.

[2]Note that in the experiments, we flipped the coin for every token. We also tried to flip the coin once per sequence, but the results were much worse, most probably because consecutive errors are amplified during the first rounds of training.

[3]In fact, in the experiments we report in this paper, our proposed approach was not meaningfully slower (nor faster) to train than the baseline.

[4]https://catalog.ldc.upenn.edu/LDC93S1.

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
