[Reviews · NeurIPS 2015]

Submitted by Assigned_Reviewer_1

Recent work on neural machine translation and other text generation tasks has trained models directly to minimize perplexity/negative log-likelihood of observed sequences. While this has shown very promising results, the setup ignores the fact that in practice the model is conditioning on generated symbols as opposed to gold symbols, and may therefore be conditioning on contexts that are quite different from the contexts seen in the gold data.

This paper attempts to remedy this problem with by utilizing generated sequences at training time. Instead of conditioning on the gold context it utilizes the generated context. Unfortunately at early rounds of the algorithm this produces junk, so they introduce a "scheduled sampling" approach that alternates between the two training methods based on a predefined decay schedule inspired by curriculum learning.

The strength of this paper is in its simplicity and the comprehensive empirical testing. The important model and inference assumptions are defined clearly, and the details about the internal architecture of the model are appropriately elided. It seems like it would be very straightforward to re-implement this approach on LSTM's or any other non-Markov model.

Empirically, the method seems to work quite well. There is a relatively large gain across several quite different tasks, and the schedule part seems to have a significant effect as always sampled does quite poorly.

The parsing results still pretty far behind state-of-the-art, but they use a very reduced input representation (no features). The speech results also seem to be using a somewhat unique setup, but the improvement here is quite large.

- I would be to know how performance changes based on the footnote 1. It seems like flipping on a token level is very different than flipping on an example level, since the worst-case distance between gold tokens is much lower.

The main weakness is the lack of comparison to other methods that attempt a similar goal.

For one, the authors are too quick to dismiss early-update perceptron (Collins and Roark, 2004) with beam search as being not applicable, "as the state sequence can not easily be factored". While the factoring is utilized in parsing, nothing about beam search requires this assumption to work.

(This connection between beam search and DP is also made on l.130. Beam search is rarely used for HMMs, at least in NLP, and when it is, it is often exactly when it is not possible to use dynamic programming.) The continuous nature of the state shouldn't effect the use of this algorithm, and in fact there is a paper at ACL this year "Structured Training for Neural Network Transition-Based Parsing" that that using this method on neural-net model that makes similar assumptions.

Secondly, I did not feel like an appropriate distinction was made with SEARN and reinforcement learning type algorithms. The related work talks about these as as "batch" approaches. While the SEARN paper itself may have chosen to use a batch multi-class classifier (since they are fast), that does not mean it couldn't be applied in the SGD case. It seems like the key idea of SEARN is to interpolate the current prediction from the model with gold based to produce a sampled trajectory. The major difference is that they may learn the policy versus using a schedule.

Summary: This paper clearly presents a simple method that yields improvements across several sequence modeling tasks. My only concern is that there does not seem to be any serious baseline comparisons, and other past methods are, to my mind, inappropriately dismissed as non-applicable.

Submitted by Assigned_Reviewer_2

The usual The usual approach to training recurrent nets, each prediction is all based on the current hidden state and the previous correct token (from the training set). But for test, we actually expect the trained rnn could generate the whole sequence by make prediction based on the previous self-generated token. The paper suggests during the training we should force the model to gradually generate the whole sequence (the previous token more and more likely generate by the model self).

Quality

Technically sound and the usefulness of scheduled sampling is supported well.

Clarity

The paper is well written and organized.

Significance:

The main idea is well motivated and interesting. The new training process could have important impacts on the study of recurrent net training.

Minor comments Do you have any intuition of the differences of the three decay schedules? How those different decay schedules represent on the training set? Does the training easily remain stuck in sub-optimal solution? Training recurrent nets could very tricky (there are lots of choices, momentum, gradient clipping, rmsprop and so on). Please provide more details of the training and make the experiments reproducible. Please also report the cost on the training set. Would the scheduled sampling be helpful for optimization?
Summary: It is a good paper that proposed a simple and straightforward scheduled sampling strategy for alleviating the discrepancy between training and inference of recurrent nets applied on generating sequences. The trained recurrent nets by this scheduled sampling outperform some fairly strong baselines on image captioning, constituency parsing and speech recognition.

Submitted by Assigned_Reviewer_3

TL;DR This paper describes a training heuristic, scheduled sampling (SS), for RNNs in which gold truth labels are sometimes replaced with sampled predictions from the model. Different schedules for deciding when to replace the gold labels are suggested, which all amount to different kinds of decay functions (linear, exponential, inverse sigmoid). Improvements over a comparable RNN model without SS are presented for the following tasks: image captioning, constituency parsing, and speech recognition.

This is a neat experimental result! While noise injection is an old idea, the focus on improving robustness at test time is interesting. But I worry that this paper raises more questions than it answers. Here are some specific concerns:

- If SS is working as a regularizer, it's good to know that it appears to be additive to dropout. However, it would also have been good to include the following baseline: what about *always* randomly sampling a label (according to the proposed schedules) rather than using model predictions?

- If the idea is to mitigate search error, I would have liked to see a comparison to baselines which use different beam widths. Is there still a benefit from SS if the model uses a larger beam width?

- I'm a little worried about the hyper-parameter k. Setting it based on "expected speed of convergence" is a little nebulous, as there's no discussion of how sensitive it is, or how it was tuned in the experiments.

Aside from these specific concerns, at a high level I think this paper would benefit from a more rigorous probabilistic analysis. It would be great if the paper shed some light on *why* the proposed heuristic appears to work, e.g. by teasing apart the regularization effect. I would have liked to see some experiments showing the benefit of SS as the amount of supervision is varied.

UPDATE AFTER AUTHOR RESPONSE:

Thanks for addressing some of my concerns. However, I still worry a little bit about how difficult it is in practice to tune the sampling schedule, and wish there a little more analysis of the method.
Summary: This paper describes a neat training heuristic for RNNs that improves robustness of predictions at test time. While the reported experiments are encouraging, it's not clear why the proposed method works and there's a severe lack of analysis (both experimental and theoretical).

Author Feedback
Author rebuttal: We thank the reviewers and will try to improve the document according to their comments.

R1:
-What about sampling once per sentence? We will experiment on this sampling scenario, and you are right that the schedule will have to be different.

-Comparison to Collins and Roark? We agree that one can use beam search with recurrent nets (we actually use it during decoding at test time). We meant that with continuous state models, there is no dynamic programming (DP) approach, so the effective number of paths considered in the search space is equal to the beam size, while it is much bigger with Markovian models that can benefit from DP. Since it takes about a week to train an image captioning model, a beam of 10 used during training would mean about 10 weeks to train the model.

-Why not compare with SEARN since it can also use online training? We didn't mean that the model in SEARN needed to be trained with a batch algorithm, but rather than there was an additional level of training, where each iteration  trained a new model to completion according to a given policy. Since it takes about a week to train one model for image captioning, such an outer loop becomes prohibitive. Designing an online version of SEARN where only one model would be trained and the policy would be updated after every single example might be possible, but is very different from what was proposed and analyzed.

R2:
-Which decay schedule to use and is training stuck in local optima? Empirically, we have observed that it is usually better to start training for a while with a very high level of epsilon_i (use-the-truth), and then decrease it with some schedule. The actual schedule is task dependent, and we selected it with a separate validation set. If we start with a low level of epsilon_i (and hence sample more from a poorly trained model), training becomes much harder and often never converges.

-Provide more details about training recurrent nets? We will add more details about the chosen hyper-parameters, but note that they were all selected on the baseline model, and only epsilon_i was tuned (on the validation set) for the scheduled sampling models.

-What about the training error? It was much higher when using scheduled sampling, but that was expected since the task in itself became more difficult. It is thus hard to compare these numbers.

R3:
-What about always random sampling a label? This is an interesting suggestion in order to show that the proposed approach does not only inject noise but that the "noise" follows the current behavior of the model, and so training slowly goes towards a test scenario, which pure noise would not provide. We will launch a few experiments on these baselines and report on them in the final version.

-Was Dropout tuned? It was tuned on a separate but related task (pentreebank) with no scheduled sampling. The best level, 30%-drop, was selected and kept for this paper. Dropout was used between input embeddings and recurrent states, and between recurrent states and output embeddings, but not between recurrent states.

-If it was about mitigating search error, did you vary the beam size? We think our approach is not just to mitigate search error, but rather to teach the model to better behave in situations where it will likely fall at test time with less guidance. Regarding the beam size, we were also expecting to need a smaller beam but we did not observe it, probably because there is some misalignment between the optimized loss (log likelihood) and the metric (say, BLEU for image captioning) and beam size was selected according to the metric on the validation set.

-How was k selected (the schedule)? k is task dependent, and thus needs to be selected according to a validation set. We think it reflects somehow the complexity of the task (and the number of updates it would take to train a baseline model to completion)

-Experiments showing the benefit of SS as the amount of supervision is varied? We could add a table where we vary the schedule from more aggressive to less aggressive and show how it impacts performance. For instance in the captioning experiments, as the schedule becomes more aggressive, performance improves for a while, and then deteriorates: CIDER started with 0.895, then 0.9, 0.901, 0.907, 0.921, and then abruptly down to 0.896 and worse. So there is an optimal schedule that is task dependent.

-Add more references about noise injection? Thank you for all these references. We will add them in the final version.

R6:
-More theoretical justification would strengthen the paper. Current recurrent models are trained in a different regime than they are used at test time, so most classical analyses about expected generalization errors do not hold. We think that our approach is a step in reducing the difference between the training and test distributions, hence getting better estimate of the generalization performance.